# Genetic Testing for Neonatal Respiratory Disease

**DOI:** 10.3390/children8030216

**Published:** 2021-03-11

**Authors:** Lawrence M. Nogee, Rita M. Ryan

**Affiliations:** 1Eudowood Neonatal Pulmonary Division, Department of Pediatrics, Johns Hopkins University School of Medicine, Baltimore, MD 21287, USA; 2Division of Neonatology, Department of Pediatrics, Case Western Reserve University, Cleveland, OH 44106, USA; Rita.Ryan@UHhospitals.org

**Keywords:** respiratory distress syndrome, pulmonary surfactant, interstitial lung disease, persistent pulmonary hypertension of the newborn, primary ciliary dyskinesia

## Abstract

Genetic mechanisms are now recognized as rare causes of neonatal lung disease. Genes potentially responsible for neonatal lung disease include those encoding proteins important in surfactant function and metabolism, transcription factors important in lung development, proteins involved in ciliary assembly and function, and various other structural and immune regulation genes. The phenotypes of infants with genetic causes of neonatal lung disease may have some features that are difficult to distinguish clinically from more common, reversible causes of lung disease, and from each other. Multigene panels are now available that can allow for a specific diagnosis, providing important information for treatment and prognosis. This review discusses genes in which abnormalities are known to cause neonatal lung disease and their associated phenotypes, and advantages and limitations of genetic testing.

## 1. Introduction

Respiratory disease, often in association with prematurity, remains the most common reason for admission to a neonatal intensive care unit. The most common neonatal lung disease associated with prematurity is respiratory distress syndrome (RDS), caused by deficient production of pulmonary surfactant owing to lung immaturity. Surfactant is a complex mixture of lipids and proteins needed to prevent end-expiratory atelectasis. Its production increases with advancing gestation, which explains the susceptibility of premature infants to RDS. Although primarily a disease of preterm infants, RDS also occurs in near-term and full-term infants, and genetic mechanisms disrupting surfactant metabolism can result in diffuse lung disease in near-term and full-term infants that mimics RDS in prematurely born infants. Persistent pulmonary hypertension of the newborn (PPHN) may complicate the course of infants with RDS, as well as other parenchymal lung diseases, including meconium aspiration syndrome, congenital pneumonia, and pulmonary hypoplasia. Pulmonary hypertension and abnormalities of lung development may also arise from genetic mechanisms. It can be difficult to distinguish infants with RDS and PPHN who will improve with supportive care from those with genetic causes, which often portend a poorer prognosis. This review will focus on identifying infants with potential genetic causes of neonatal lung disease, indications for genetic testing, and the strengths and limitations of different approaches to genetic diagnosis. To determine which infants should have genetic testing, it is first necessary to review the common genetic disorders that typically lead to respiratory distress in the newborn period. Note that during this review we avoid the use of the term “mutation” in favor of “sequence variation.” Strictly, mutation means change, but in common parlance has become associated with a change that causes disease, which is not necessarily the case.

### 1.1. Genetic Causes of Diffuse Lung Disease/Respiratory Distress Syndrome

Although the phospholipid components of surfactant, particularly disaturated phosphatidyl choline (DSPC), are critical for its ability to lower surface tension, specific proteins have important roles in surfactant function and metabolism [1]. Not surprisingly, DNA sequence variations in the genes encoding these proteins can eliminate or reduce their function, resulting in a phenotype of RDS ([2] for review). A summary of the specific genes and proteins is listed in Table 1 and key features are summarized.

Surfactant Proteins B (SP-B) and C (SP-C) are low-molecular-weight extremely hydrophobic proteins that enhance the surface tension lowering properties of surfactant lipids, particularly their ability to adsorb to and spread at the air–liquid interface. SP-B and SP-C are critical components of commercially available replacement surfactants routinely used to treat premature infants with RDS. They are encoded by separate genes (called *SFTPB* and *SFTPC*) located on human chromosomes 2 and 8, respectively. Proteolytic processing of much larger precursors, called proSP-B and proSPC, generates the mature forms of SP-B and SP-C found in the airspaces.

Given their role in surfactant function, it is not surprising that genetic mechanisms disrupting or altering the production of SP-B and SP-C might result in lung disease. Complete loss-of-function sequence variants on both alleles of *SFTPB* cause severe, diffuse lung disease that is usually present at birth. SP-B deficiency is progressive and usually fatal within the first three months of life, although rare infants have been reported who have survived into childhood owing to sequence variants that allow for some SP-B to be produced. As sequence variations are needed on both alleles for disease, the disease is inherited in an autosomal recessive fashion [5,6].

Sequence variations in *SFTPC* also cause diffuse lung disease, but in contrast to the loss-of-function sequence variants in *SFTPB*, the sequence variations in *SFTPC* lead to production of an abnormal form of proSP-C. The abnormal proSP-C is injurious to the cell and represents a toxic gain-of-function mechanism. Although some infants present at birth with an RDS-like picture, the majority of patients with sequence variations in *SFTPC* present later in life, ranging from early infancy to well into adulthood. A typical presentation is an infant who presents with failure to thrive, hypoxemia, tachypnea and other signs of respiratory distress, and diffuse interstitial and/or alveolar infiltrates on chest imaging. In contrast to SP-B deficiency, sequence variants on just one allele are usually sufficient to cause disease, which are thus inherited in an autosomal dominant pattern with variable penetrance, or cause apparent sporadic disease due to de novo sequence variations in *SFTPC* [7,8,9,10].

Pulmonary surfactant is stored in specialized organelles in alveolar type 2 cells called lamellar bodies, prior to its secretion into the airspaces. Lipids critical for surface tension lowering are imported into lamellar bodies by member A3 of the adenosine triphosphate (ATP) binding cassette family of membrane transporters, ABCA3. Sequence variants in the gene (*ABCA3*) disrupting or limiting production of ABCA3 may therefore cause surfactant deficiency [11]. The clinical picture is often similar to that of SP-B deficiency, with early-onset RDS in a full-term infant that is progressive and fatal within several months of life. The nature of the sequence variant (genotype) influences the phenotype. Infants with complete loss-of-function sequence variants on both alleles invariably present shortly after birth and have a fatal course [12]. These are usually nonsense or frameshift variants predicted to completely preclude any ABCA3 production. Infants with other kinds of sequence variants (missense, splice-site, small in-frame insertions or deletions) have a much more variable course. Though some infants have severe, early-onset and ultimately fatal disease, others have a milder course, and may not even have lung disease at birth [13,14,15]. Presumably, sequence variants associated with milder disease allow for some functional ABCA3 to be produced [16,17]. The exact amount of functional ABCA3 needed to prevent lung disease is not known. ABCA3 deficiency is inherited in an autosomal recessive fashion with variants needed on both alleles to manifest disease. However, some infants with sequence variants on just one allele (monoallelic, single, or heterozygous) may sometimes develop lung disease, particularly if they are born prematurely or near-term [18]. ABCA3 production is developmentally regulated, increasing with advancing gestation, and lung disease resulting from monoallelic sequence variants is likely due to the combination of decreased production or function arising from one allele owing to the sequence variant, combined with reduced expression due to immaturity, resulting in a level of ABCA3 production or function below that needed to prevent lung disease. As ABCA3 expression may continue to increase with maturity, such infants usually recover from their lung disease, although this has not been formally studied.

In addition to the hydrophobic proteins SP-B and SP-C, surfactant contains two larger, structurally related hydrophilic proteins, SP-A and SP-D, which are encoded in a multigene complex on chromosome 10. There are two genes for SP-A, *SFTPA1* and *SFTPA2*, that encode similar but distinct proteins, SP-A1 and SP-A2, and a single gene for SP-D (*SFTPD*). Both proteins have a collagenous domain and a carbohydrate binding or lectin-like domain, and are thus part of the collectin family, important proteins in the innate immune system. SP-A and SP-D have important immunomodulatory roles and roles in innate immunity in the lungs, helping to protect against a wide range of bacterial and viral pathogens, which is likely their main role in surfactant. Deletion of the genes encoding SP-A and/or SP-D does not result in neonatal lung disease in experimental animals, nor have sequence variants in these genes been reported as causes of neonatal lung disease. Missense sequence variants in the genes encoding SP-A have been identified in adults with pulmonary fibrosis, often in association with lung cancer, but very rarely in children [19,20,21,22]. Sequence variants in *SFTPD* have not yet been reported as a cause of human disease [22].

The regulation of expression of the surfactant system during development is complex, but thyroid transcription factor 1 (TTF−1) is important for the expression of ABCA3 and the surfactant proteins. As the name implies, TTF−1 is also important for thyroid gland development, and it is also expressed in other tissues, including the central nervous system. Sequence variants in or deletions of the gene for TTF−1, *NKX2–1*, can cause lung disease, ranging from severe neonatal respiratory failure to milder disease later in life [23,24]. As *NKX2–1* is important in other organ system development, hypothyroidism and neurological symptoms may often be present, with the full spectrum referred to as “brain-thyroid-lung” syndrome. However, abnormalities in all organ systems need not be present in all patients, and some do not have any lung disease. Based upon limited data, the nature and severity of the lung disease may depend upon which downstream genes are most affected. For example, decreased expression of ABCA3 or SP-B results in more severe, neonatal onset disease, whereas alterations in SP-C expression result in interstitial lung disease (ILD) in older individuals [23]. Decreased expression of SP-A and SP-D may result in susceptibility to infection or severe lung disease following viral infection [24,25]. *NKX2–1* is also important more globally in lung development, which may be impaired in patients with abnormalities in the gene [26]. The mechanism for disease appears to be inadequate production of TTF−1 due to an abnormality on one allele (haploinsufficiency). Thus, disease may be inherited as an autosomal dominant with variable penetrance, or sporadic disease may occur due to de novo sequence variants. The majority of, but not all, neonates have some evidence of thyroid dysfunction. Neurological symptoms are highly variable, with movement disorders (chorea, ataxia) most prominent in older individuals. Neurologic findings may be absent or very subtle in newborns, and also may be attributed to the severity of the lung disease.

### 1.2. Genetic Causes of Altered Lung Development and Pulmonary Hypertension

Hypoxemic respiratory failure with diffuse lung disease radiographically is a characteristic finding in children with SP-B deficiency and the severe forms of ABCA3 deficiency. Pulmonary hypertension may also be present. A similar clinical picture can occur in children with primary disorders of lung development. Alveolar capillary dysplasia with misalignment of the pulmonary veins (ACD-MPV) is a disorder characterized by abnormal pulmonary vascular development. A hallmark feature is the presence of pulmonary veins in the same bronchovascular bundle adjacent to pulmonary arteries, the so-called misalignment of the veins, although more recent work suggests that these vessels represent abnormal or persistent anastomotic connections [27]. Affected children typically present shortly after birth with severe hypoxemia and PPHN. The course is usually progressive and unresponsive to pulmonary vasodilator therapies as well as support from extracorporeal membrane oxygenation, although infants with relatively milder disease are being increasingly recognized [28,29]. Anomalies of other organ systems, including cardiac, genitourinary and gastrointestinal systems, are common [30].

The principal cause of ACD-MPV results from deletions or sequence variants in one allele of the gene encoding the transcription factor, FOXF1 [30,31]. The majority of cases appear to arise sporadically from de novo sequence variations, although dominant inheritance with variable penetrance has been observed. Deletions in the untranslated region upstream of the gene encoding two long noncoding RNAs (lncRNAs) important for FOXF1 regulation have also resulted in the phenotype of ACD-MPV [32,33,34]. Children with a consistent clinical and pathological phenotype but no identifiable variants in *FOXF1* have been recognized, so other genes may also be involved, but have yet to be identified [33].

Acinar dysplasia is a rare disorder characterized by interruption of lung development in the late pseudoglandular to early canalicular phase, precluding acinar development. Congenital alveolar dysplasia is an even rarer disorder characterized by arrest of lung development at the late canalicular or early saccular stage [35]. Infants with acinar dysplasia or congenital alveolar dysplasia have severe respiratory failure at birth that is refractory to interventions, similar to those with ACD-MPV or severe surfactant deficiency. Recently, sequence variations on one allele of the gene encoding the transcription factor TBX4 were identified in infants with lung pathology consistent with acinar dysplasia [36]. Deletions of or sequence variants in this gene are also a cause of pulmonary hypertension in older infants and children, and may also be associated with skeletal abnormalities [37]. The genetics of acinar dysplasia and congenital alveolar dysplasia are complex. Genic and intragenic deletions or sequence variants in TBX4, as well as genes encoding components of the signaling pathway downstream of TBX4, including FGF10 and FGFR2, have been identified in affected infants [38]. Heterozygous variants in other genes, including *ABCA3*, were also found in some infants, so the precise phenotype may depend upon interactions from multiple genes [39].

### 1.3. Genetic Causes of Transient Neonatal Respiratory Distress

Genetic disorders disrupting surfactant function, metabolism, or lung development cause lung disease that is usually progressive. However, sequence variants in *SFTPC* may cause severe lung disease early in life, including in the newborn period. However, with supportive care infants with *SFTPC* variants may stabilize and even improve with time, although their lung disease usually does not resolve completely. As described above, newborns with monoallelic *ABCA3* sequence variants may have an RDS phenotype at birth, particularly if born prematurely, but subsequently improve and even apparently resolve their lung disease. However, the natural history and long-term pulmonary outcomes for such children have not been formally studied. Finally, with increased availability of genetic testing, less severe cases of what were previously believed to be uniformly neonatal lethal disorders are being increasingly recognized [12,15,28,29,40].

In addition, upwards of 80% of children with primary ciliary dyskinesia, a disorder of mucociliary clearance, present in the newborn period with respiratory distress and require neonatal intensive care [41,42,43,44]. These children are more likely to have onset of symptoms >12 h, lobar infiltrates and a need for supplemental oxygen that persists for >48 h, with the majority diagnosed as neonatal pneumonia [45]. As only ~50% of children with primary ciliary dyskinesia (PCD) have abnormal situs, the disease may not be suspected in the neonatal period. The mean age of diagnosis of children with PCD was 4.7 years in a prospective study, suggesting that an opportunity for early diagnosis is being missed [41,44]. An argument can be made that genetic testing is warranted in all neonates with respiratory distress persisting for >48 h that is not adequately explained by history and other laboratory findings yielding a specific diagnosis [46]. PCD results from DNA sequence variants in genes responsible for ciliary assembly and function. Currently, almost 40 genes in which variants may cause the phenotype of PCD have been identified, but this number is likely to grow. Electron microscopy of ciliary structure from a nasal mucosal biopsy can be helpful for diagnosis, but some genes are associated with normal ciliary ultrastructure [47,48]. A full review of PCD is beyond the scope of this review, but is discussed in detail in a separate review in this journal (https://www.mdpi.com/2227–9067/8/2/153, accessed on 9 March 2021).

### 1.4. Other Genes Responsible for Neonatal Lung Disease

Variants in other genes have been recognized to result in neonatal lung disease, of which the phenotypic features may overlap with those resulting from sequence variants in surfactant-related or lung developmental genes. The phenotypes of these children often involve extrapulmonary manifestations. Examples include structural genes including filamin A, a cytoskeletal protein. Genetic variants in this gene, *FNLA*, which is located on the X-chromosome, also result in cardiac, skeletal and brain abnormalities (periventricular heterotopias) [49]. The lung disease is characterized by hyperinflation and may present in the neonatal period. In prematurely born infants, it may resemble severe bronchopulmonary dysplasia [50,51]. Its inheritance is as an X-linked dominant disorder and hence it is more prevalent in females, although affected male infants have been reported. Several other genetic disorders associated with neonatal onset lung disease are listed in Table 2, recognizing that this list is not all inclusive.

### 1.5. Practical Issues of Genetic Testing for Neonatal Lung Disease

The disorders discussed above are all rare, but associated with significant morbidity and mortality. It can be difficult to distinguish these disorders from more common causes of neonatal lung disease, such as RDS, that respond well to available therapies and generally have a good prognosis. Lung biopsy can be helpful in establishing a diagnosis, but is invasive and associated with potential increased morbidity in an already critically ill child. Moreover, it may not be specific—while there are characteristic features indicative of a genetic surfactant disorder, those characteristics do not indicate which gene is causative, which is important for prognosis. Genetic testing can be done on a blood or saliva sample and thus has the advantages that it is non-invasive as well as potentially yielding a specific diagnosis. Disadvantages include that testing may be expensive, costs may not be paid by insurance, and it can often take considerable time for results to be returned (turn-around time). It can also be challenging to decide which infants warrant genetic testing, when to send it, and what tests to send as genetic testing methods are constantly improving. Finally, the interpretation of results may not be straightforward. The major problem with interpretation is sequence variants of which the clinical significance is unknown (termed variants of unknown significance, or VUS). These are generally single base changes that are predicted to result in a change of a single amino acid in the protein, but of which the effect on the metabolism or function of the protein is unknown. Determining the potential significance of a VUS can be challenging. Factors to be considered include the nature of the amino acid change and its location in protein, evolutionary conservation, and population frequencies of the identified variant from publicly available databases from exome and genome sequencing projects such as the Genome Aggregation Database (gnomAD; gnomad.broadinstitute.org, accessed on 9 Mar 2021). Although there are a number of in silico programs to predict pathogenicity, they may yield conflicting results and are predictions only. Family segregation studies may be helpful—appropriate segregation of the variant(s) with the phenotype in multiple affected individuals or de novo occurrence may support the potential pathogenicity of a given variant. Guidelines for interpretation of DNA sequence variants have been published [55].

The methods used for genetic testing initially relied on targeted sequencing of selected genes based on the clinical presentation (phenotype) using standard Sanger sequencing methods. This method involves direct amplification of portions of the gene using the polymerase chain reaction and then direct sequencing of the fragments. Although highly sensitive for single nucleotide changes and small insertions or duplications, it is not very cost effective as the sequence data from each reaction are limited in size and to a portion of one gene, so multiple reactions need to be performed individually to fully cover one gene or multiple genes. Sanger sequencing is also not sensitive to large deletions or duplications of one allele, as the opposite intact allele would be sufficient to generate sequence data and thus separate analytical methods are needed if such a deletion or duplication (copy number variant, or CNV) is suspected.

Next generation sequencing (NGS) methods employ chemical sequencing of multiple small fragments of a gene or genes simultaneously, and then using computer algorithms to align the fragments of sequence data to the reference sequences. The average number of times a base or region of a gene is sequenced is referred to as the depth of coverage, and important for the sensitivity of the approach for detecting sequence variants. For a targeted panel of genes, an enrichment step is used to amplify or capture the genes of interest and improve coverage for those genes, enhancing the sensitivity. In addition, by comparing the coverage depth across a gene it is possible to determine whether there is extra or reduced sequence data from one allele, thus making this approach able to detect larger deletions and duplications. A major advantage of NGS is that multiple genes may be sequenced simultaneously, reducing the costs of sequencing and genetic testing substantially, and allowing for the design of multigene panels based on a phenotypic presentation that could result from sequence variations in different genes. Whole exome sequencing (WES) uses enrichment strategies and NGS methods to cover almost the entire coding sequence (exons), which represent 1–2% of the human genome. Whole genome sequencing (WGS) approaches analyze almost the entire human genome, including non-coding regions. The potential feasibility of WGS approaches for very rapid diagnosis of suspected neonatal genetic disorders has been demonstrated, but requires sophisticated analytical methods to determine the potential importance of the very large number of variants from reference sequences identified in a given individual—>30,000 for WES, and 4,00,000–5,000,000 for WGS. One caveat is that as such methods rely on sequencing relatively small fragments of DNA (50–200 bases), disorders that arise from expansion of repeat sequences (congenital myotonic dystrophy, fragile X syndrome) require different approaches. There are now multiple clinical diagnostic labs offering NGS panels of multiple genes for both pulmonary and other disorders, with varying costs and turn-around times ranging from 14 days to 8 weeks. These panels may be based on an exome platform, meaning that WES is performed, but only the data for the genes of interest are extracted. The advantage of such an approach is that if additional genes are suspected that were not initially requested, or new genes are discovered that may result in the phenotype of interest, the relevant data for that patient simply need to be extracted, rather than performing additional sequencing, reducing costs and time for results. An additional benefit is that such approaches may identify variants in genes not yet known to be associated with the phenotype, but which are plausible candidates based on their function, such as those encoding for mucins or non-motile cilia. A searchable reference for different labs offering genetic testing may be found at www.ncbi.nlm.nih.gov/gtr (accessed on 9 Mar 2021). Table 3 lists which genes or gene families are most likely based on the clinical presentation (phenotype). A family history of neonatal lung disease, diffuse lung disease in older children or young adults, or consanguinity should prompt early investigation.

Although NGS approaches allow for cost- and time-effective analyses of multiple genes, a disadvantage can be the amount of data returned and their interpretation. The more genes that are included in a panel, the higher the likelihood of identifying VUSs in one or multiple genes, often ones in which there was an a priori low suspicion for causing the phenotype. This is particularly true for WES or WGS, but when the clinical presentation presents a broad differential diagnosis, studying multiple genes may yield an unsuspected diagnosis that would not have been detected by more targeted testing. Though WES and WGS studies are expensive, they can be cost-effective for critically ill infants in intensive care units with complex or extreme phenotypes for which there is a broad differential diagnosis [56,57,58,59]. As WES and WGS are agnostic approaches, they also have the potential advantage of revealing possible interactions between different genes in determining a phenotype, as demonstrated in studies on the genetics of lethal lung developmental disorders [38]. An important limitation is that our current understanding of the impact of DNA sequence variants on phenotypes is limited, particularly with respect to variants in untranslated regions, which would not be identified by most approaches other than WGS. Table 4 lists advantages and disadvantages of different genetic testing methodologies (adapted from [60]).

### 1.6. Candidates for Genetic Testing

Determining which babies warrant genetic testing for rare disorders and when to send such testing can be challenging. Many of the disorders considered portend a poor prognosis, and timely referral for lung transplantation may be the only therapeutic option. There are also considerable economic, emotional and other costs from sustaining such infants in intensive units, requiring multiple invasive procedures that may ultimately prove futile. As it may take weeks for results of genetic testing to return, a high index of suspicion based on knowledge of their typical presentations is essential. A family history of lung disease should prompt suspicion for a genetic mechanism, but this is often absent for recessive disorders, and sporadic disease may arise from de novo sequence variations. As a general statement, such disorders should be considered when the presence and/or severity of disease does not seem to be explained by the clinical history. As an example, a 40-week-gestation female infant delivered vaginally after an uncomplicated pregnancy and delivery who develops a clinical picture consistent with RDS in a preterm infant should be a candidate for early testing. In contrast, with a late preterm male infant delivered operatively to a mother with gestational diabetes who has several recognized risk factors for RDS, it may be reasonable to defer testing while seeing how the baby responds to supportive care. In general, if an infant has persistent lung disease for which there is not a good explanation, consultation with genetics should be considered earlier rather than later. Depending upon the laboratory and approach, a definitive diagnosis based on genetic testing may be available in 2−3 days [58,61,62].

Selective use of genetic testing has a potentially high yield. Figure 1 shows the results of analyses from a 20+ year study designed to identify infants with inborn errors of surfactant metabolism (. Nogee, unpublished data). Criteria for entry included gestational age > 35 weeks, diffuse lung disease radiographically requiring invasive ventilator support, and no clear explanation for the lung disease based on clinical history. Children with younger gestations or milder disease were eligible if there was a family history of neonatal lung disease or interstitial lung disease in a parent. Overall, 44% had a proven or likely genetic mechanism identified.

## 2. Conclusions

Genetic mechanisms are rare but important causes of both transient and progressive neonatal lung disease. Understanding the typical clinical presentations of infants with such disorders is critical to establishing a diagnosis, which can be done non-invasively through genetic testing of a blood, saliva, or buccal sample. Such testing is readily available through multiple laboratories, but has limitations in terms of cost, turn-around time, and difficulties in interpretation. Collaboration with an engaged geneticist is critical in arriving at a diagnosis in the most efficient manner. Families of infants with positive results should also be referred for formal genetic counseling.

## Figures and Tables

**Figure 1 children-08-00216-f001:**
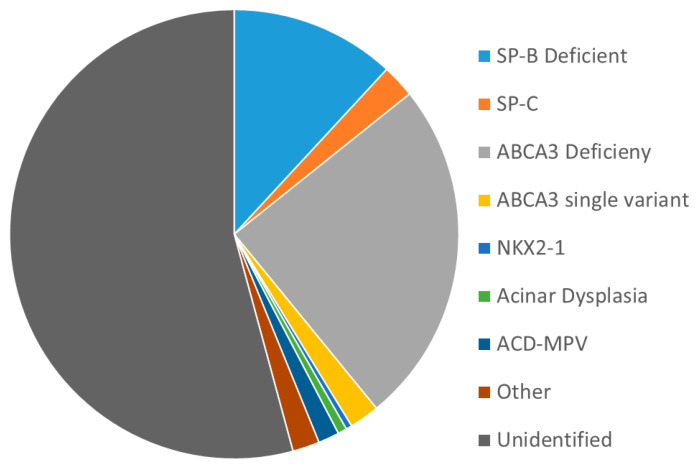
Analysis of 463 infants with severe, diffuse neonatal lung disease, 1995–2016. Analyses were focused on surfactant-related genes. Complete analyses could not be performed on all subjects as the number of genes to be analyzed included in the consent evolved over time, and retrospective analyses for genes not included in more limited consents were not performed, or insufficient samples remained. A genetic mechanism was identified in 44% of subjects.

**Table 1 children-08-00216-t001:** Genetic surfactant related disorders.

Protein	SP-B	SP-C	ABCA3	SP-A	SP-D	TTF−1	GM-CSF Receptor [3,4]
Gene	*SFTPB*	*SFTPC*	*ABCA3*	*SFTPA1* *SFTPA2*	*SFTPD*	*NKX2–1*	*CSFR2A* *CSFR2B*
Pulmonary Phenotypes	RDS	ILD PF RDS	RDS PPHN ILD PF	PF Lung cancer	None yet known	RDS ILD Recurrent Infection	Alveolar Proteinosis
Inheritance	AR	AD sporadic	AR	AD sporadic	N.A.	Sporadic AD	AR
Prognosis	Rapidly fatal	Variable	~60% rapidly fatal; ~40% variable	Generally adult onset, progressive	N.A.	Variable	Childhood to adult onset; variable
Incidence	<1 in 1,000,000	Unknown	Uncertain, 1 in 10 K to 1 in 20 K	Unknown	N.A.	Unknown	Unknown

RDS, respiratory distress syndrome; ILD, interstitial lung disease; PF, pulmonary fibrosis; PPHN, persistent pulmonary hypertension of the newborn; AR, autosomal recessive; AD, autosomal dominant.

**Table 2 children-08-00216-t002:** Pulmonary phenotypes, related genes and extrapulmonary findings.

Category	Clinical Phenotype(s)	Genes	Extrapulmonary
Diffuse Lung Disease/ Surfactant Dysfunction	RDS PPHN	*ABCA3* *SFTPB* *SFTPC* *NKX2–1*	Isolated Lung Disease Hypothyroidism and neurological abnormalities
Pulmonary Hypertension Pulmonary Hypoplasia	PPHN	*FOXF1* *TBX4*, *FGF10* *TMEM70* [52]	Multiple other organ systems Skeletal abnormalities PAH in older children Hypertrophic cardiomyopathy Hypotonia Hyperammonemia
Primary Ciliary Dyskinesia	Respiratory Distress Congenital Pneumonia “Wet” cough (in infancy)	Multiple	Situs Inversus Situs Ambiguous
Other	RDS Hyperinflation, “BPD” RDS	*ITGA3* [53] *FNLA* (Filamin A) *TMEM170*	Skin and Renal CNS heterotopias Cardiac and Skeletal Skin and joint, immune dysfunction, SAVI (“STING-associated vasculitis of infancy”) [54]

RDS, respiratory distress syndrome; PPHN, persistent pulmonary hypertension of the newborn; PAH, pulmonary arterial hypertension; BPD, bronchopulmonary dysplasia; STING: stimulator of interferon genes.

**Table 3 children-08-00216-t003:** Likely genetic mechanisms depending on neonatal clinical presentation.

Clinical Phenotype	Factors Prompting Suspicion for Genetic Cause	Possible Genes	Additional Clinical Features
RDS, full-term	Lack of risk factors for RDS including: preterm gestation (<36 weeks) IDM, operative delivery without labor, clinical suspicion for infection.	*SFTPB* *ABCA3* *NKX2–1* *SFTPC*	*NKX2–1* may be associated with hypothyroidism, neurological symptoms
RDS, preterm	Severity and/or persistence of disease out of proportion to that expected for infant’s gestational age or clinical history (SGA, likelihood of congenital pneumonia)	*ABCA3* *FLNA*	Associated with monoallelic variants Hyperinflated CXR
PPHN	Lack of risk factors for PPHN (MAS, perinatal depression) or pulmonary hypoplasia (severe oligo or anhydramnios due to obstructive uropathy or early, prolonged ROM)	*FOXF1* *TBX4, FGF10*	Extrapulmonary manifestations common
Unexplained respiratory distress	Onset > 12 h Need for supplemental oxygen >48 h Lobar Infiltrates	PCD genes	~50% with abnormal situs
Family history of neonatal lung disease, diffuse lung disease in older children or young adults, consanguinity should prompt early investigation			

RDS, respiratory distress syndrome; SGA, small for gestational age; PPHN, persistent pulmonary hypertension of the newborn; ROM, rupture of membranes; PCD, primary ciliary dyskinesia; PCD, primary ciliary dyskinesia.

**Table 4 children-08-00216-t004:** Advantages and disadvantages of different genetic sequencing approaches.

Method	Advantages	Disadvantages
Sanger Sequencing	Long reads (up to 1000 bp) Very sensitive to SNV Relatively rapid Sensitive to small in/dels, repeat expansions	Relatively expensive Targeted to a single gene Will not detect CNVs
NGS panels (e.g., “surfactant protein gene panel” or “congenital hypotonia gene panel”)	Cost effective Can be targeted to genes of highest likelihood based upon phenotype Can be designed to detect CNVs Considerable heterogeneity in which genes covered, cost, TAT depending upon specific laboratory	Short reads (50–200 bp) Prespecified genes dependent on enrichment step Limited coverage of untranslated regions Turn-around times longer Identified variants require confirmation by sanger sequencing Likelihood of detecting VUSs in genes with lower probability of being responsible for disease
WES	Cost-effective sequencing of all known coding regions for detection of SNVs Rapid TAT possible May detect variants in unanticipated genes Potential for identification of novel genes (research)	Short reads Enrichment/capture step needed Not all genes/regions well covered Higher cost than NGS panels Increased number of VUS identified Incidental findings *
WGS	Cost-effective sequencing of almost entire genome, including untranslated regions Can be faster than WES Rapid TAT feasible Potential for identification of novel genes (research)	Variable coverage of some genes/regions Less sensitive to non-SNV changes Higher cost than WES Informatics more complicated VUS will be detected Incidental findings *

SNV, single nucleotide variation; in/del, insertion or deletion; CNV, copy number variant; TAT, turn-around time; NGS, next-generation sequencing; WES, whole exome sequencing; WGS, whole genome sequencing; VUS, variant of unknown significance. * refers to detection of pathogenic or likely-pathogenic variants unrelated to phenotype being investigated, but with clinical significance. (e.g., tumor susceptibility, neurodegenerative diseases).

## Data Availability

The data presented in this study are available on request from the corresponding author. The data are not publicly available due to privacy.

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
