# Peer review of "Genetic Testing for Neonatal Respiratory Disease"

_children, 2021, doi:10.3390/children8030216_

Round 1

Reviewer 1 Report

This review presents the reader with a succinct overview of the known genes involved in NRDS. Overall the narrative and scientific understanding is excellent. The main focus is on the surfactant genes, which are known to cause of NRDS and are relatively well described in the literature. NRDS is thought to occur in 80% of PCD births but the genes that cause PCD are not discussed in any great detail. The number of genes that cause PCD now exceeds 40. This section could be expanded?

The main problem I have is that the title reads like there would be advice for the reader as to what to do for gene testing individuals with NRDS. The “who, why, what, when, how”, part should be removed because the authors do not answer in any detail many of these questions.

Who- No genetic evidence is stated as to the likely ethnicities of those affected by NRDS. The consanguineous inheritance effect or other reasons for genetic variation in individuals may be more relevant to who gets tested?

Why- There is an argument that all neonates with NRDS should be genetically tested as stated here: Late Diagnosis of Infants with PCD and Neonatal Respiratory Distress. Goutaki et al, J Clin Med. 2020 Sep 4;9(9):2871

What- Do the authors mean what genes to test or what disease to test for? Very confusing and I could not find any guidance is in the body of the manuscript.

When-The timing for genetic testing is not explained in any detail for the different types of gene variants.

How- There is no detailed discussion on how the genetic tests should be done for each gene. For example NGS is used in some centres and gene panels can be out of date and likely to miss any newly discovered gene variants not on the panel. WES may be more accessible in certain centres and may become the test of choice-does this warrant more discussion? The other point is that genetic testing is limited by our genetic knowledge, and it does not address intronic DNA changes so variation in the regulators of gene expression will be overlooked.

A table comparing the available tests and advantages, disadvantages, cost etc would enhance the paper.

Mucin gene variants are also likely to cause NRDS but no gene panel exists for testing these, this should be mentioned.

More non-motile ciliopathies are presenting with respiratory symptoms, could these cases be missed in NRDS? And should we test these gene panels?

Grammatical errors:

  • Line 92- ABCA3 in may therefore -> ABCA3 may therefore
  • Line 189- as well genes encoding -> as well as genes encoding
  • Line 321- “As WES and WGS approached are agnostic approaches” doesn’t make sense.

Reviewer 2 Report

I truly enjoyed this concise review of the numerous genetic causes for neonatal respiratory disease when common-place surfactant insufficiency of prematurity (+/- maternal diabetes) RDS, meconium aspiration, infection, etc. have been ruled out. The discussion of the available gene array platforms and indications was well received. While technical, the style and flow of writing was easy to follow even for a non-geneticist. I think this review will engage readers of multiple disciplines and levels of expertise.

Minor edits and comments are listed below to polish this already quality review.

  1. Abstract Line 11: recognized [as] rare causes
  2. Line 41: Why was neonatal underlined? Was this intentional to delineate from pediatric lung disease?
  3. Line 66: “and pro-SPC…” Should be proSP-C
  4. Line 92: editing needed “disrupting or limiting production of ABCA3 in may therefore cause”
  5. Table 1: overall easy to follow and arranged well; recommend reformatting in the order discussed in the text; also see comment #14 below
  6. Line 201: “particularly if often born prematurely…” remove “often”
  7. Line 206: “upwards of 80% [of] children with primary ciliary dyskinesia (PCD)”
  8. Line 215: “Currently almost 40 genes in which variants may cause the phenotype of PCD [have been identified], but this number is likely to grow.”
  9. Line 223: “Genetic variants in this gene, (FNLA), which…” parentheses not needed
  10. Table 2: the gene and extrapulmonary manifestation columns have not maintained their alignment in my version, please check that they line up the way intended. Also, typo in space parenthesis orientation: “Wet” cough( in infancy)
  11. Line 257: “…such as such as the Genome Aggregation” delete duplicate “such as”
  12. Line 288: can you clarify better what the 1-2% refers to; presumably 1-2% is the uncovered portion? Reads oddly.
  13. Figure: remarkable that nearly half of the suspected unexplained/atypical lung disease had a genetic diagnosis! I am presuming that “463” is the number of subjects - as currently labeled/bold format it appears to be the Figure #.
  14. Several genes/disorders are included in tables (CSFR2, TMEM70 and 170(?), NPC2, ITGA3) but not discussed in the text; if not added to the text it would be helpful if the authors could include any recommend references for these genes not discussed to guide interested readers in their search.
